# Socioeconomic Inequalities as a Cause of Health Inequities in Spain: A Scoping Review

**DOI:** 10.3390/healthcare11233035

**Published:** 2023-11-24

**Authors:** Guillem Blasco-Palau, Jara Prades-Serrano, Víctor M. González-Chordá

**Affiliations:** 1Centro de Salud Museros, Conselleria de Sanitat Universal i Salut Pública, Generalitat Valenciana, 46136 Museros, Spain; 2Centro de Salud Pintor Sorolla, Conselleria de Sanitat Universal i Salut Pública, Generalitat Valenciana, 46010 Valencia, Spain; jaraprades@gmail.com; 3Nursing Research Group (GIENF-241), Universitat Jaume I, 12006 Castelló de la Plana, Spain; vchorda@uji.es; 4Nursing and Healthcare Research Unit (Investén), Institute of Health Carlos III, 28029 Madrid, Spain

**Keywords:** scoping review, public health, health inequality, socioeconomic status, income, universal coverage, social determinants of health, Spain

## Abstract

The objectives of this review were to identify the population groups most frequently studied, to determine the methods and techniques most commonly used to show health inequities, and to identify the most frequent socioeconomic and health indicators used in the studies on health inequities due to socioeconomic inequalities that have been carried out on the Spanish healthcare system. A scoping review was carried out of the studies conducted in the Spanish State and published in literature since 2004, after the publication of the Law of Cohesion and Quality of the National Health System. The PRISMA extension for scoping reviews was followed. The methodological quality of the studies was assessed using the critical reading guides of the Joanna Briggs Institute and an adaptation of the STROBE guide for ecological studies. A total of 58 articles out of 811 articles were included. Most of the articles were (77.59%, *n* = 45) cross-sectional studies, followed by ecological studies (13.8%, *n* = 8). The population group used was uneven, while the main geographical area under investigation was the whole state (51.7%, *n* = 30) compared to other territorial distributions (48.3%, *n* = 28). The studies used a multitude of health and socioeconomic indicators, highlighting self-perception of health (31.03%, *n* = 19) and social class (50%, *n* = 29). The relationship between better health and better socioeconomic status is evident. However, there is variability in the populations, methods, and indicators used to study health equity in Spain. Future health research and policies require greater systematization by public institutions and greater cooperation among researchers from disciplines such as sociology, economics, and health.

## 1. Introduction

Inequality in health is a reality throughout the world, which has reached alarming gradients in certain parts of the globe. However, in Europe, where most countries have health systems that, in general, can be considered to be universal, there are also health inequities perpetuated over time despite increased quality of life [1,2]. In such national health systems, financing comes from the general budgets of the States and the States supervise the provision of services, so that all citizens have access without distinction to all the services offered, without added payments. 

In this situation, social class is one of the most important social distribution systems since its multiple aspects comprehensively affect all aspects of society. With regard to Marxist origins, Marx himself did not create a systematic definition of the term social class, originating multiple debates throughout history. However, Marx identified social class depending on the place occupied with respect to the means of production, divided between the bourgeoisie and the proletariat, although he also included the petty bourgeoisie or the lumpenproletariat (a group of socially marginalized people, sometimes performing illegal work or not contributing anything socially) [3]. Other authors have developed exhaustive studies and definitions of social class, and have attemped to update this concept from a relational perspective, incorporating differentiators such as control over monetary capital, the physical means of production, or the labor force based on supervision and discipline. Thus, Wright proposed a map with twelve class positions, with their contradictions and complex exploitations, and differentiated the twelve types based on those who own and those who are salaried, and then among the different levels of existing hierarchies, which made the map more operational [4]. For his part, Bordieu [5] explained that social class could be defined by the structure of the relationships among different properties, with each property granted its own value according to the effects it exerts on the others.

In this labyrinth of definitions, the British Register General has attempted to synthesize the concept to give it practicality by drawing up a social classification of six steps according to occupation, as follows: (I) professionals and managers; (II) intermediate charges; (IIIa) non-manual workers; (IIIb) skilled manual workers; (IV) semi-skilled manual workers; (V) unskilled manual workers. Likewise, this classification has been used in previous studies on social inequalities and health inequities [6,7] and presents sufficient validity for the present study.

At the end of the 20th century, researchers became interested in social inequalities and the consolidation of the different universal health systems in Europe. In 1980, a group of English researchers published the book Health Inequalities, later known as the “Black Report” due to the name of its main researcher [8]. As McIntosh [8] related, in this study, an attempt was made to make visible, among many other things, that even in health systems with universal coverage, clear health inequalities have been observed and, as they argued, have been due to class inequities. This research has initiated interest in health inequalities among modern researchers, serving as the basis for the design of new studies on health inequities. Other international actors have joined in and have periodically conducted their own research on this subject, including the World Bank and the World Health Organization [9].

Over the years, different types of health inequality studies have emerged, each focused on a cause and consequences, or the relationship between them, and various analytical methods have been used, leading to the development of disparate theories. Thus, Mackenbach [10] synthesized these investigations into nine theories according to the possible causes and consequences. The author based part of his analysis on the paradox that with clear improvements in the different states of well-being there is an increase in health inequities in countries with high incomes. Some of these theories include those described by Marmot [11] and Wilkinson (cited by [8]) in which they relate inequality to psychosocial stress derived from deprivation for socioeconomic reasons, arguing with data the different exposures to this stress. Likewise, Batty [12] and Mackenbach [13] related socioeconomic position to personality and cognitive development, considering that they also affect health largely reducing the external factors to the individual in the inequities, quite contrary to the other theories. In contrast, Lynch and Davey Smith, cited by Mackenbach [10], focused on health inequality due to material deprivation of people, which results in unequal exposure to different life experiences.

Specifically, Spain has had a national health system since 1986, which is characterized by universal coverage, public financing, and quality benefits in a decentralized system in autonomous communities [14]. In 2003, the Law on Cohesion and Quality of the National Health System [15] was promulgated to guarantee citizen participation, as well as quality and equity of care in the national territory. In 2008, the Commission for the Reduction of Social Inequalities in Health was established. This Commission proposed 166 recommendations, divided into 14 areas, and declared the need for a commitment to promote health and equity in all policies and to move towards a fairer society [16]. The Spanish national health system is considered to be one of the best health systems in the world based on analyses of global data [17,18], but differences in outcome indicators have been observed when data from autonomous communities are compared [19,20]. In addition, some studies have tried to synthesize the literature on health inequities [21], but from broader perspectives and considering, in addition to social class, determinants such as gender, age, or the type of territory, as they can have a multiplying effect on health inequities [16]. Therefore, the objectives of this review were to identify the population groups most frequently studied, to determine the most commonly used methods and techniques to show health inequities, and to identify the most frequent socioeconomic and health indicators used in studies on health inequities due to socioeconomic inequalities that have been carried out in the Spanish State since the publication of the Law of Cohesion and Quality of the National Health System.

## 2. Materials and Methods

### 2.1. Design

A scoping review was carried out following the methodological framework proposed by Arksey and O’Malley [22] and Levac et al. [23], in which a review process was carried out divided into the following five stages: (i) identifying the research question; (ii) identifying relevant studies; (iii) selecting studies; (iv) charting the data; (v) collating, summarizing, and reporting the results. Likewise, we followed the recommendations of the “Preferred Reporting Items for Systematic Reviews and Meta-analysis extension for Scoping Review (PRISMA-ScR)” [24].

### 2.2. Phase I: Identifying the Research Question

One of the purposes of a scoping review is to explore the variety of material available on a specific research topic [22]. Therefore, it was considered to be an adequate method to address the following research question: What types of inequity, populations, methods, and indicators have been used in the research on health inequalities due to socioeconomic causes, which has been carried out in Spain since the publication of the Law on Health Cohesion and Quality of the National Health System?

### 2.3. Phase II: Identifying Relevant Studies

The bibliographic search was carried out in pairs in the electronic databases PubMed, Web Of Science (WOS), and Scopus. Similarly, the investigation was completed in the following three virtual libraries: Cochrane Library, Scientific Electronic Library Online (SciELO), and the Virtual Health Library (BVSalud). The databases and virtual libraries were chosen for their international recognition and broad multidisciplinary coverage, with the intention of recovering articles relevant to the topic of the review. A search strategy was established by combining keywords extracted from the structured vocabularies Descriptors in Health Sciences (DeCS) and Medical Subject Headings (MeSH), as well as natural language. Table 1 shows the general strategy used for the different bibliographic databases, with the necessary adaptations.

Two reviewers (G.B.P. and J.P.S.) were confirmed to have the same results after searching each database. The search strategy was applied to each database chosen according to the possibilities they offered, trying to maintain homogeneity. In addition, the snowball technique was used in the bibliographic reference lists of the articles included in the review to identify articles that had escaped the search strategy.

### 2.4. Phase III: Study Selection

All studies with data collected from 2004 and up to the time of the bibliographic search (14 March 2023), whose field of research was Spain and which were written in Spanish, Catalan, and English, were included. The design types included were descriptive observational studies, including ecological and cross-sectional studies, and analytical observational studies, such as case-control and cohort studies. The publication of the Law on Cohesion and Quality of the National Health System [15] was considered to be a starting point, which tried to modernize, expand, and consolidate the health rights already developed in the General Health Law [14], establishing social participation, quality, and equity as common rights to all citizens. Articles that did not fall within the objectives set out in the review and those that did not have a research article format, such as letters to the editor, editorials, thematic updates, or similar, were excluded.

The studies to be included in the review were selected in pairs following the selection criteria previously described. First, each reviewer (G.B.P. and J.P.S.) independently read the title and abstract of each article to make a first selection. Second, each reviewer independently performed a full reading of the selected papers to confirm if they met the selection criteria and to perform a critical appraisal of the methodological quality of the studies. For this, the critical reading guide for cross-sectional studies was used, as well as cohorts and control cases, proposed by the Joanna Briggs Institute [25]; the STROBE checklist adapted to ecological studies by Dufault and Klar [26] was also used. Two reviewers (G.B.P. and J.P.S.) classified the articles according to their methodological quality into three levels, i.e., good, average, or poor. Thus, it was necessary for the two reviewers (G.B.P. and J.P.S.) to assign, at least, the average category to the same article. The results of both reviewers were compared, and the research group discussed discrepancies until a consensus was reached. This procedure was established since the instruments used to assess methodological quality did not establish cut-off points to determine the level of methodological quality.

### 2.5. Phase IV: Charting the Data

Data extraction was performed by preparing a table that included information on the researchers involved in the study, the year, period, type of study, target population, health and socioeconomic indicators, main results, and methodological quality. Data extraction was carried out separately by the two researchers (G.B.P. and J.P.S.), who later compared the results and discussed them to ensure the uniformity and consistency of the data.

### 2.6. Phase V: Collating, Summarizing, and Reporting the Results

First, a descriptive analysis (frequencies and percentages) of the search results was carried out according to the year and type of publication, as well as the methodological quality. Then, the articles were descriptively analyzed according to the specific objectives of the review to determine the target populations and the most commonly used health and socioeconomic indicators. Specifically, health and socioeconomic indicators were grouped into 14 and 11 categories, respectively, due to the great variability among the articles included in the review.

## 3. Results

A total of 811 articles were retrieved and, among them, 278 articles were removed since they were duplicates. Figure 1 presents the review flowchart. A total of 1.5% (*n* = 12) of the articles were retrieved from the Virtual Health Library, 2.6% (*n* = 21) of the articles were retrieved from the Web Of Science, 8.4% (*n* = 68) of the articles were retrieved from the SciELO (Scientific Electronic Library Online), and 38.7% (*n* = 314) of the articles were retrieved from Pubmed; Scopus was the largest database, with 48.5% (*n* = 393) of the articles. Next, two researchers (G.B.P. and J.P.S.) collated the reports and excluded 34.3% of the articles (*n* = 278). Subsequently, the selection criteria were applied to the remaining 65.7% (*n* = 533) of the articles. Among these 533 articles, 20.3% (*n* = 108) of the articles were excluded for exceeding the proposed time limits. Another 23.6% (*n* = 126) of the articles were excluded for exceeding the Spanish geographic location, 1.3% (*n* = 7) of the articles were excluded for not using indicators, 1.9% (*n* = 10) of the articles were excluded due to the size of the population, 8.4% (*n* = 45) of the articles were excluded for having a format that was not relevant for their inclusion, and 44% (*n* = 235) of the articles were excluded for not being focused on the subject of study. Finally, after the exclusion process, there were a total of 64 (12%) articles that respected the proposed selection criteria. It is necessary to clarify that the gap produced between the percentages is because the same article could fail to meet different criteria simultaneously.

In the last stage of the selection process, an evaluation of the methodological quality was carried out. For example, concerning the article by Abellán et al. [27], the scales used and the levels of studies evaluated were not standardized, regularly used, or comparable, including their income distribution, limiting the study to only dichotomous variables. In the case of Fernández-Martínez et al. [28], despite being an interesting article in some aspects, it was not adapted to the purpose of this review, and, among other reasons, we rejected it for using economic self-perception since it is too subjective an indicator. Following the corresponding checklists, 9.4% (*n* = 6) of the 64 articles included were excluded because they did not meet the minimum quality standards. Finally, 67.2% (*n* = 43) of the articles were assessed as having good methodological quality, and the remaining 23.4% (*n* = 15) of the articles as acceptable. Thus, a total of 58 studies were included in this scoping review.

A total of 77.6% (*n* = 45) of the included articles were cross-sectional studies, followed by ecological studies (13.8%, *n* = 8), cohort studies (5.2%, *n* = 3), case-control studies (1.7%, *n* = 1), and longitudinal studies (1.7%, *n* = 1). The time period was variable, including studies in which only a single moment, an annuity, or even up to four years of data were analyzed. In addition, variability was observed in the target population. Some studies focused on a specific population with certain diseases and on how some socioeconomic variables (education, social class, or place of residence) affected the course of the ailments (10.34%, *n* = 6). In other studies, specific population segments were used, such as children, the elderly, or young people (44.83%, *n* = 26), while the rest (*n* = 26, 44.83%) included the general population. Likewise, the geographical area was also not homogeneous, with 51.7% (*n* = 30) of the articles focusing on the entire Spanish State, while the remaining 48.3% (*n* = 28) of the articles used different geographical distributions, such as autonomous communities, provinces, cities, or even neighborhoods.

Table 2 shows information on the health and socioeconomic indicators. On the one hand, the health indicators related to chronic diseases or comorbidities (31.03%, *n* = 18), self-perception of health (32.76%, *n* = 19), behavioral factors or lifestyles (24.14%, *n* = 14), and mental health (24.14%, *n* = 14) were the most used categories. Health indicators related to unsatisfied health needs (3.45%, *n* = 2), life expectancy (3.45%, *n* = 2), and environmental pollutants (1.72%, *n* = 1) were the least used. On the other hand, socioeconomic indicators related to educational level (58.62%, *n* = 34), social class (50%, *n* = 29), income and purchasing power (36.21%, *n* = 21), and area of residence and urban planning (24.14%; *n* = 14) were the most used, while poverty was only present in one article (1.72%, *n* = 1).

Appendix A offers a complete list of references included in the review and the extraction of the information from the articles can be found in Table 3, including information on the authors, objective, design, study population, study period, health and socioeconomic indicators, results, and assessment of the methodological quality. Regarding the population, not previously mentioned, it was observed that the different researchers used data from primary sources, not having been collected by themselves. In addition, 36.2% (*n* = 21) of the articles used complete samples since they were obtained from official bodies that had all this information, while the remaining articles, i.e., 63.8% (*n* = 37), used projections from surveys or other sources of population data. Regarding the years used to extract the data, we identified three situations. A total of 20.69% (*n* = 12) of the studies were carried out prior to 2008, while 29.31% (*n* = 17) of the studies were carried out during the economic crisis, comparing before and after the recession. Finally, 50% (*n* = 27) of the studies considered the information after the crisis, due to the interest it aroused. In addition, all the articles’ analyses corroborated the existence of health inequalities, although with greater or lesser intensity and focusing on different aspects. Moreover, a complete list of references included in this review is available in the Appendix A.

## 4. Discussion

As we proposed in the objectives, we found that cross-sectional studies are the most frequent design for analyses of health inequities. Furthermore, we observed that there was variability in the populations studied, although, importantly, the studies were carried out on the general population and the availability of data may have influenced this aspect. Finally, the multiple indicators used at the socioeconomic and health levels were grouped, showing a clear picture of the most frequent indicators, i.e., self-perception in health and educational level as the most frequent indicators, respectively.

We found that there are abundant studies in the literature on health inequalities, although very heterogeneous and disintegrated, without forming a body of their own. In this sense, the studies carried out in the Spanish State affirmed the presence of inequities in health due to different factors related to social stratification and also showed that the existence of a national health system is not a sufficient condition for eliminating these inequalities, which are very present throughout Spain.

There has been an increase in research on social inequalities and their effects on health inequities, coming from various perspectives. Likewise, the involvement of different organizations not specific to the study of these data, such as the International Monetary Fund [86] and La Caixa Foundation [87], confirms the increased interest in studying equity in health. In addition, the deep economic crisis experienced since 2008, the creation of the Commission on the Reduction of Social Inequalities in Health in 2010, and the recent SARS-CoV-2 pandemic have given rise to many new investigations on health inequities.

However, although there has been an increase in the number of studies on health inequities, there is a lack of standardization in terms of methodology or indicators, despite the efforts of some authors such as Ruiz-Alvarez et al. [21]. Thus, the authors’ experiences or interests within their respective disciplines have seemed to guide the research process on inequalities and inequities in health, but not so much that the research has been approached as a subject in which to specialize [88]. This implies that many researchers have studied health inequalities sporadically without developing a broad literature on the matter. In addition, this isolated interest means that the use of different indicators, depending on the appeal at the specific moment, have hindered the methodological standardization required to rigorously address the study of health inequity.

The majority of the studies have been cross-sectional studies. After reading the studies, it was possible to infer some of the possible reasons, such as the use of secondary data from the National Health Survey [89] or other sources at the state or regional level such as the Catalan Institute of Statistics (IDESCAT) [90]. To use these data, researchers have resorted to the microdata provided by surveys to relate them as they wish, analyzing specific people at an exact moment. With few exceptions, the general trend of data recovery and subsequent analysis has seemed to follow this method. Precisely, the source of the data used has determined the indicators used. Thus, both socioeconomic and health indicators have depended on the organism, limiting creativity. Likewise, the availability of data from different institutions has not always been sufficient; sometimes, data have not been comparable, or their disaggregation has not allowed the development of synthetic indicators. In this regard, the studies analyzed showed unfavorable results. In general, there is a relationship between a lower disposition of economic or intellectual capital and a greater inequality in health. At this point, therefore, the question arises as to why this occurs in a country like Spain, where health coverage is universal and free. There are different explanations for this and the studies analyzed present some possible hypotheses. For example, in a survey of the use of health services by patients with HIV [54], it is stated that some barriers have prevented the poorest people from equitably demanding these services, even when they are free. Taking into consideration that, in most cases, these are self-reported data, Álvarez-Gálvez et al. [42] argued, in one of their points, the importance of the mediating effect of perceived discrimination and happiness on health. The use of these subjective indicators by the majority could suggest the impact that collective psychology, hope for the future, and territory could have on self-perceptions, and therefore on their results. However, practically all the studies have shown that a lower socioeconomic status was associated with worse health, whatever the approach and indicators used.

As discussed, there are multiple approaches to inequality in health. Although collective approaches that emphasize social causes are traditionally dominate, studies that put the individual and their different characteristics at the center of their research are increasing. Following the trend in other fields of study where the decisions, capacities, or more immediate contexts determine the results in health, relating personality or cognitive ability to socioeconomic inequality leads to health inequities [13]. Thus, on the one hand, the social dynamics that relegate large pockets of the population to exclusion are left in the background, ignoring the importance of marked socioeconomic differences. On the other hand, we found that the Black Report, as argued in a subsequent analysis, emphasized the importance of social over individual indicators and reduced the explanatory magnitude of individual lifestyles in these health inequities [91]. In this regard, there is a point of friction between the different researchers that should be considered.

Despite not being the main objective of the study, other racial or gender inequalities should be discussed since they increase inequities in health, acting in some cases as inequality multipliers. Ruiz Álvarez et al. [21] addressed this aspect in another review from an intersectionality perspective, while this study aimed to present a class approach, understanding that even within race or gender, class oppression operates in a very pronounced way. In other words, it is understood that even internally, these categories are not homogeneous and present divergences for socioeconomic reasons; therefore, although most of the studies considered them due to their multiplier effect, an attempt was made to separate those in which they were the main object of study. Nevertheless, both reviews use similar methodologies, which allows them to reach similar conclusions regarding the type of articles and indicators used, although both works can largely complement each other.

This scoping review has some limitations in its implementation, such as the spatial limitation of the article, as it focuses exclusively on Spain and may have left out multiple studies with a great contribution to the field in question. There may also be an evaluation bias that can always be found, despite trying to minimize it by performing a peer review. In addition, selection bias may have left out relevant studies or, finally, may have been incurred publication bias by considering the published studies, tending to positive results in the proposed hypotheses. The establishment of the descriptors and the choice of the databases could have also caused a bias when obtaining the studies. Despite these limitations, the results of this study are of interest to decision makers and researchers interested in social inequalities and inequities in health since they show the need to standardize the methods and indicators to rigorously approach their study.

## 5. Conclusions

The results of this literature review show that, in the literature, there has been an increase in studies carried out in Spain related to socioeconomic inequalities and a lack of equity in health, especially during the deepest years of the crisis. In general, the research has been focused on cross-sectional studies that correlate socioeconomic inequalities and health inequity in the general population, although we also observed ecological, cohort, or case-control studies. Meanwhile, the most widely used health and socio-sanitary indicators are self-perceptions of health and educational level, respectively. Other indicators are less prevalent, although with a rather notable presence in the different studies, such as chronic diseases, lifestyle, or mental health on the health side and social class or income as socioeconomic indicators.

The results of the analyzed studies should alert public officials to the importance of addressing socioeconomic inequalities in order to reduce existing health inequities and not only focus on partial measures or increasing health spending as the only solution. However, the methodological characteristics described surely limit the impact that these investigations can have on political decision making. Future studies are needed that focus on methodological consensus to address the study of this problem and to develop and monitor composite indices that allow an objective and adequate assessment of the impact of policies aimed at reducing social inequalities and inequity in health.

## Figures and Tables

**Figure 1 healthcare-11-03035-f001:**
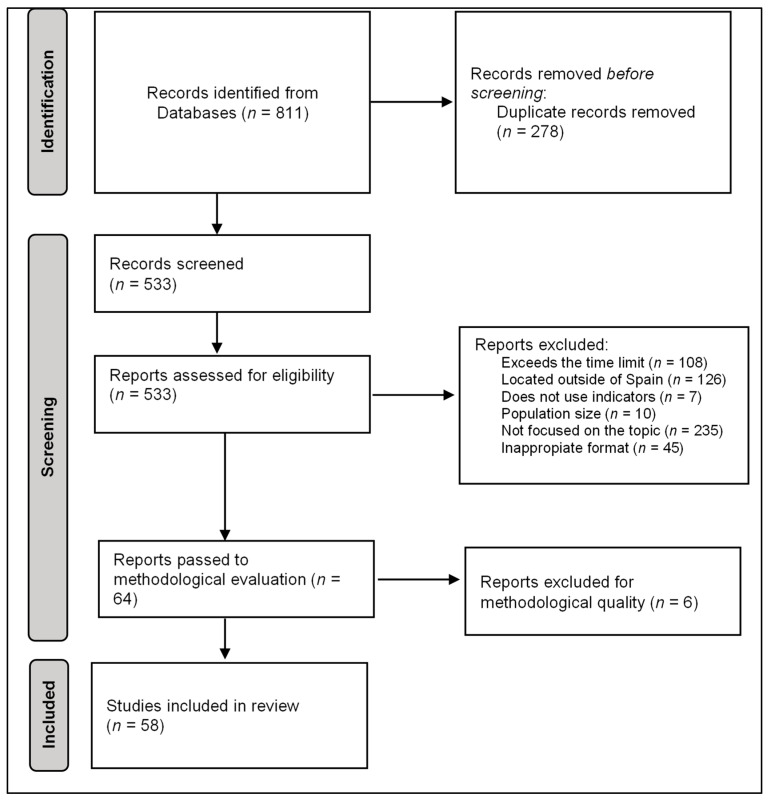
Review flowchart.

**Table 1 healthcare-11-03035-t001:** General outline of the search strategy.

((“inequalities”[All Fields] OR “inequality”[All Fields] OR “inequities”[All Fields] OR “inequity”[All Fields]) AND (“health”[MeSH Terms] OR “health”[All Fields] OR “health”[All Fields] OR “healthful”[All Fields] OR “healthfulness”[All Fields] OR “healths”[All Fields]) AND (“socioeconomic factors”[MeSH Terms] OR (“socioeconomic”[All Fields] AND “factors”[All Fields]) OR “socioeconomic factors”[All Fields] OR (“socioeconomic”[All Fields] AND “factor”[All Fields]) OR “socioeconomic factor”[All Fields]) AND (“spain”[MeSH Terms] OR “spain”[All Fields] OR “spain s”[All Fields])) AND ((catalan[Filter] OR english[Filter] OR spanish[Filter]) AND (2003:2020[pdat]))

**Table 2 healthcare-11-03035-t002:** List of health and socioeconomic indicators.

Health Indicators	% (*n*)	Socioeconomic Indicators	% (*n*)
Self-perception of health	32.76% (19)	Educational level	58.62% (34)
Chronic diseases/comorbidity	31.03% (18)	Social class (job occupation)	50.00% (29)
Behavioral factors/lifestyles	24.14% (14)	Income and purchasing power	36.21% (21)
Mental health	24.14% (14)	Urbanism, area of residence *	24.14% (14)
Mortality	13.79% (8)	Composite indices **	22.41% (13)
Medical visits/use of the health system/ Deficiencies in the system (waiting lists)	12.07% (7)	Unemployment status	22.41% (13)
Disability/difficulties on activities of daily living	10.34% (6)	Social support and household composition	20.69% (12)
Secondary prevention	6.90% (4)	Nationality	18.97% (11)
Acute diseases	5.17% (3)	Social exclusion/perceived discrimination	6.90% (4)
Maternity/paternity characteristics	5.17% (3)	Life satisfaction (self-perception)	6.90% (4)
Life expectancy	3.45% (2)	Poverty	1.72% (1)
Unmet health needs	3.45% (2)	-	-
Health expenses	1.72% (1)	-	-
Environmental pollutants	1.72% (1)	-	-

* Household type, place of residence, property value, home renovations, new construction homes, urban/rural, homes with no heat or bathroom, areas in need. ** Social exclusion determinant index, deprivation index, Foster–Greer–Thorbecke (FGT) index, vulnerability index, job insecurity scale, area-based deprivation index.

**Table 3 healthcare-11-03035-t003:** Extraction of information.

Authors	Objective	Design	Population	Time Period	Health Indicators	Socioeconomic Indicators	Main Results	Quality
Compés Dea, M.L. et al., (2018) [29]	To evaluate the use of the Population and Housing Census_2011 and develop a deprivation index by basic healthcare area (BHA) and to analyze its association with mortality in Aragon.	Ecological	Population of Aragon on 1 November 2011, residing in main housing, *n* = 1,131,189	2011–2014	Standardized mortality rate (SMR)	Insufficient education, insufficient education for young people, insufficient education for 16–64 year olds, foreigners insufficient education, single-parent households for mothers, single-parent households for fathers, single-person households for over 65 years, elderly, elderly, foreigners, manual workers, unemployment, casual wage earners, unheated housing, housing without a bath, housing without a shower, housing without internet, small dwellings, dwellings with outstanding payments, dwellings for rent, buildings not accessible, buildings without a lift, buildings in poor condition, houses not accessible, houses without a lift, substandard housing	Socioeconomic inequalities detected with deprivation index are relevant in mortality.	Good
Esteban Peña, M.M. et al., (2020) [30]	To determine the level of association between territory and vulnerability, specifying proposals for territorial intervention using key socio-health indicators.	Cross-sectional	Non-institutional population of the city of Madrid, *n* = 9513	2017	Self-perception of health status, quality of life in relation to health, tobacco consumption, alcohol consumption, self-occurrence of obesity, sedentary, mental health	Vulnerability index	The results support the behavior of global and specific health indicators with vulnerability, with a disaggregation by sex, which will allow planning adapted to the territory.	Good
Aguilar-Palacio, I. et al., (2016) [31]	To know the use of primary care (PC), specialized care (SC), hospitalizations, day and emergency hospitals, and hyperfrequentation in the elderly in Spain, analyzing the influence of health status, sex, social class, and temporal evolution.	Cross-sectional	Spanish population over 65 years old, *n* = 13,613 (63.3% women)	2006–2012	Consultations with PC and SC in the last 4 weeks, hospitalization and day hospital in the last year, and use of emergency in the last year. Perceived health in the last 12 months	Social class	The elderly of low social classes used more frequently AP and emergency, while the use of AE and day hospital was greater in upper classes.	Good
Tornero Patricio, S. et al., (2017) [32]	To analyze the effect of the place of housing in pediatric hospitalizations and describe the hospitalization rates of the most frequent major diagnoses in the city of Seville.	Cross-sectional	Children under 15 living in the city of Seville, *n* = 2660 (43.23% girls)	1 January 2014 and 31 December 2014	Type of admission, hospital, hospital stay, discharge type, and main diagnosis	Postal code, district, and ZNTS (areas in need of social transformation)	Hospitalization rates for the most frequent major diagnoses were higher in districts with lower socioeconomic status. Their income was lower and more urgent.	Average
Rajmil, L. et al., (2010) [33]	To analyze the socioeconomic and health factors associated with child mental health in a representative sample of the population aged 4–14 in Catalonia.	Cross-sectional	Children under 14 in Catalonia, *n* = 1821	December 2005 and July 2006	Mental health, and stated health problems (from a list of common problems in childhood)	The highest occupational social class of the family, the educational level of the mother, the type of family	Children from families with lower socioeconomic status and single-parent families are at risk of worse mental health.	Good
Saez, M. et al., (2019) [34]	To confirm the effect environmental problems have on socioeconomic inequalities in health by using intra-urban geographical areas as the units of analysis, given that these are already mostly clustered by socioeconomic conditions.	Ecological	Inhabitants of the city of Barcelona, *n* = 1,604,555, (759,820 men (47.06%) and 845,035 women (52.94%))	2007–2014	Total male and female yearly mortality by neighborhood, environmental noise data, and air pollution	Disposable household income, the percentage of foreigners from low income countries, and housing prices	The risk of dying due to environmental hazards in a very affluent neighborhood is about 30% lower than in a very depressed neighborhood. The effect of environmental hazards was more harmful to the residents of Barcelona’s most deprived neighborhoods,	Good
Bilal, U. et al., (2018) [35]	To study the associations between neighborhood socioeconomic status and diabetes prevalence, incidence, and control, in the entire population of northeastern Madrid, Spain.	Ecological	Population over 40 years old living in the city of Madrid, *n* = 269,942	2013–2014	Type 2 diabetes diagnosis, lack of diabetes control ((hba1c ≥ 7%) or a continuous variable (hba1c %))	Neighborhood socioeconomic status Education, wealth (average housing prices), occupation (part-time employment, temporary employment and manual occupational class), and living conditions (unemployment rate)	Diabetes prevalence, incidence, and lack of control increased with decreasing NSES, in a southern European city.	Average
Merino-Ventosa, M. et al., (2018) [36]	To explicitly estimate the degree of income-related inequality in cytology testing for Spanish women, in 2006 and 2011, by employing concentration indices and by decomposes changes in inequality in order to ascertain how the contribution of each explanatory factor has evolved over time.	Cross-sectional	Women aged 25–64, *n* = 17,330 (all women)	2006–07 and 2011–12	Cervical cytology testing, self-reported health	Education, labor status, social class, and region of residence	Income was the main determinant of inequality in cervical screening.	Good
Tamayo-Fonseca, N. et al., (2018) [37]	To analyze the association of the risk of poor mental health with various demographic, socioeconomic, health status, quality of life, and social support variables.	Cross-sectional	The whole population of the Valencian Community For 2005, *n* = 5781 (49.5% men and 50.5% women) In 2010, *n* = 3479 (48.9% men and 51.1% women)	2005–2010	Presence of a nonmental chronic disease, number of nonmental chronic diseases, presence of a disability, self-perceived quality of life	Socioeconomic level, country of birth, level of education, employment status, personal self-perceived income level, and occupational class	The prevalence of poor mental health risk increased substantially in the Valencian Community from 2005 to 2010, and several variables were closely associated with this.	Good
Nuñez, O. et al., (2020) [38]	To determine to what extent adverse stage distribution contributed to survival inequalities in a Spanish region before the implementation of a CRC screening program.	Ecological	All patients living in a region of southern Spain with CRC (colorectal cancer) diagnosed between 2004 and 2013 (Granada), *n* = 4759 (2802 men and 1957 women)	2004–2013	Age at diagnosis, site	Deprivation quintile	Socioeconomic inequalities mainly affect CRC survival in men and these inequalities were observed especially during the first year of follow-up.	Average
Solé-Auró, A. et al., (2020) [39]	To examine how the level of education limits progress in extending health life expectancy at ages 50-plus by combining measures of length (mortality) and quality of life (morbidity) for Spanish men and women.	Cross-sectional	The entire population of Spain	2012	Mortality rates at age 50, self-perceived health over previous 12 months, the global activity limitation indicator (GALI) and reports on the presence or absence of any diseases	Level of educational attainment	There was a strong association between education and overall population health.	Average
Vives, A. et al., (2011) [40]	To determine the prevalence of precarious employment in the waged and salaried workforce in Spain, to describe its distribution across social groups defined by occupational class, gender, age, and immigrant status, and to estimate the proportion of cases of poor mental health potentially attributable to employment precariousness.	Cross-sectional	Population aged 16 to 65, living in Spain, *n* = 7650	October 2004 and July 2005	Mental health	Employment precariousness, immigrant status, educational attainment, and occupational social class	Nearly 6.5 million workers in Spain were exposed to some degree of employment precariousness in the period 2004–2005, which may be detrimental for their mental health, and almost 900,000 of them were exposed to high employment precariousness.	Good
Crespo-Cebada, E. et al., (2012) [41]	To present new evidence both on the horizontal inequity in the delivery of primary health care and on the factors driving inequalities in the use of general practitioners services for a Spanish population aged 50 years and over.	Cross-sectional	Spanish non-institutionalized people aged 50 or over	2006–2007	Total number of visits to a general practitioner, long-term illness, comorbidity, limitation (the respondent has a health problem or disability that limits the kind or amount of paid work he/she can perform), mental health, restrictions in daily instrumental activities, autoperception of their health, physical activity.	Educational level, employment status, health coverage, income level, family situation, and external home help received	At equal levels of need, rich and poor elderly people are not treated equally.	Good
Álvarez-Gálvez, J. et al., (2019) [42]	(1) To identify the underlying mechanisms in the association between SES and health in Andalusia at the end of a period of economic crisis; (2) to design a confirmatory model based on previous knowledge to measure the relevance of different dimensions of socioeconomic position and ”unmet needs” (i.e., those suffered during the periods of economic crisis) to explain the incidence of negative health outcomes; and finally, (3) to provide recommendations that might guide future social and health policies aimed to reduce health inequalities in this region.	Cross-sectional	8 provinces of Andalusia, *n* = 1200	2016	Self-perception in health (SF12), perception of being affected (in health) by the economic crisis, health difficulties during the economic crisis, unmet health needs, BMI, physical activity, fruit consumption, vegetable consumption	Net household income, education, happiness, perceived discrimination, gender, age	Our results indicated that socioeconomic factors and demographics are associated with health by influencing lifestyles, socioeconomic experiences during the crisis, and personal well-being. Additionally, several remarkable contributions from the results can be highlighted: (a) Important differences in health outcomes were found in Andalusian females, (b) there was no relationship between income and lifestyles, (c) people with unmet medical needs demonstrated a higher perception of social discrimination, and (d) discrimination and happiness had a mediating effect on health.	Good
Abásolo, I. et al., (2017) [43]	To analyze whether the recent recession has altered health care utilization patterns of different income groups in Spain.	Cross-sectional	The entire population of Spain, *n* = 29,712 for 2006 and *n* = 19,935 for 2011–2012	July 2006 and December 2011	Use of health care services, self-perceived health status, the number of chronic health conditions, drug and medicine use	Net income of the individual’s family, occupation, studies, and the number of family members	The results of this research show that universality in public healthcare provision has not prevented the financial crisis from affecting some income groups more than others.	Good
Cebrecos, A. et al., (2018) [44]	To evaluate statistical and geographical stability of an area based deprivation index (ABDI) computed at different spatial scales and to study their relation with cardiovascular disease.	Cross-sectional	Madrid residents (aged 45–70), *n* = 1,446,994	2011–2014	Cardiovascular disease (CVD) prevalence	Area based deprivation index (ABDI), social class, unemployment, educational level	The correlations between deprivation and CVD at the three study scales were positive and significant, being greater as the size of the spatial unit increased. Deprivation and adjusted rates of CVD prevalence showed greater variability at the census section level than when they were added at the district level.	Average
Maynou, L. et al., (2014) [45]	To assess the effect of the economic crisis on the temporal space variation of socioeconomic inequalities in mortality in small areas in Barcelona, Spain.	Ecological	The population studied was that of those residing in the neighborhoods of Barcelona, *n* = 1,620,943, (769,819 men (47.49%) and 851,124 women (52.51%))	2005 and 2008–2011	Mortality (crude death rates)	Disposable household income and the percentage of foreigners from low income countries	Although the geographical pattern in relative risks for mortality in neighborhoods in Barcelona remained very stable between 2005 and 2011, socioeconomic inequalities in mortality at an intraurban level have surged since 2009.	Good
Rueda, S. (2012) [46]	To analyze health inequalities among older adults in Spain by adopting a conceptual framework that globally considers two dimensions of health determinants and the mediating influence of social support, taking into account individual socioeconomic position.	Cross-sectional	People aged from 65 to 85 years with no paid work living in two socioeconomically developed regions (The Basque Country, Navarra, Andalusia, and the Region of Murcia), *n* = 1602 (535 men and 1067 women)	2006	Self-rated health status, poor mental health	Educational attainment, family characteristics (type of household, living status, and relationship with the household head) and social support	These results show the importance of implementing stronger gender equity policies, as well as reducing socioeconomic inequalities among regions and strengthening social support among older adults.	Good
Barroso, C. et al., (2016) [47]	To explore differences in the effect of socioeconomic characteristics on Spaniards’ self-assessed health status, depending on the Spanish economic situation.	Longitudinal	Spanish population over 15 years of age, *n* = 50,113 (49% men and 51% women)	2006–2007 and 2011–2012	Self-assessed health (SAH)	Professional status, household’s economic situation, educational level	The effect of socioeconomic status on SAH behaves differently during a crisis and also depends on the socioeconomic status indicator considered.	Good
Cainzos-Achirica, M. et al., (2019) [48]	To evaluate the associations between SES, measured using individual annual income, health outcomes, and public healthcare resource use in congestive heart failure patients from the general population of Catalonia (Spain).	Cross-sectional	All Catalan residents alive on 1 January 2016, aged 50 years or older and covered by the public healthcare system, *n* = 7,638,524	2016	Life expectancy, mortality, chronic heart failure	Individual annual income	Lower individual income was independently and robustly associated with higher mortality.	Good
Morteruel, M. et al., (2018) [49]	To describe inequalities in the use of nursing services, medical services in primary care, specialist care, and services not fully covered by the Basque public health system in Spain.	Cross-sectional	Population living in the Basque Country, *n* = 10,454 (5585 women and 4869 men)	2013	Any visit to general practitioners, nursing services, specialists (all three covered services), dentists, physiotherapists (not fully covered services), and podiatrists	Occupational social class and educational level	There is inequality in the use of health systems, especially in areas not covered by the national health system.	Good
Arrospide, A. et al., (2019) [50]	To identify inequalities on health by analyzing the interactive effects of gender, age, educational level, social class, body mass index, and chronic diseases on health-related quality of life in the Spanish population.	Cross-sectional	The entire population of Spain, *n* = 18,450 (51.3% were women)	2011–2012	Body mass index, chronic conditions, mobility, self-care, usual activities, pain/discomfort, and anxiety/depression	Educational level and social class	The joint analysis of gender, age, educational level, social class, body mass index, and chronic diseases on health-related quality of life in the Spanish population revealed important inequalities in health.	Average
Rocha, K. et al., (2015) [51]	To analyze inequalities in the prevalence of poor mental health and their association with socioeconomic variables and with the care network in the Autonomous Communities in Spain.	Cross-sectional	Resident population in non-institutional Spain, over 16 years old, *n* = 29,476	Between June 2006 and June 2007	Mental health (General Health Questionnaire-12)	Occupational social class, health coverage, country of origin, and employment status	The results showed that there are inequalities in the prevalence of poor mental health in Spain, associated to contextual variables, such as unemployment rate. In addition, it was observed that inequalities in the mental health care resources in the Autonomous Communities also have an impact on poor mental health.	Average
Arias-de la Torre, J. et al., (2016) [52]	(a) To determine the prevalence of poor mental health in the working population of Spain in 2011; (b) to identify the association of this prevalence with socioeconomic and work-related variables for men and women separately; (c) to determine if the patterns differ by gender.	Cross-sectional	Spanish working population between 16 and 65 years of age, *n* = 7396	2011	Mental health (with the General Health Questionnaire (General Health Questionnaire-12))	Marital status, education and training, occupational social class, type of contract, job stress, Job satisfaction	The obtained results show that the prevalence of poor mental health in the working population of Spain in 2011 was closely related to gender, being higher among women than among men. The results also provide further evidence in favor of the hypothesis that support that the socioeconomic variables could have a higher weight in the mental health of women and those related to the paid work in men.	Good
Martín, U. et al., (2012) [53]	To analyze health inequalities according to the place of birth (indigenous population and born in other autonomous communities).	Cross-sectional	Non-institutionalized population aged 50 to 79 in Catalonia 2006 *n* = 5483 (2642 men and 2841 women) and in the Basque Country 2007 *n* = 3424 (1596 men and 1828 women)	2006–2007	Self-assessment of the health	Place of birth (Autonomous Community), social class, and educational level	In both communities, there are health inequalities to the detriment of the population from the rest of Spain.	Good
García-Goñi, M. et al., (2015) [54]	To contribute to the literature by developing a different analysis looking at the first mentioned reality: Human immunodeficiency virus (HIV) health policy and planning in developed countries where access to ART is guaranteed.	Cross-sectional	People with HIV in the Basque Country, *n* = 2,262,698 (50.9% female)	2010–2011	Diagnosis of HIV, initial expenditure on health (primary care, specialized care, emergencies, rehabilitation, external, laboratory, radiology, and others such as dialysis or chemotherapy)	Rate of deprivation (dividing population by quintiles)	Equity in health provision for HIV patients represents a challenge even if access to treatment is guaranteed. Lack of information in poorer individuals might lead to underprovision, while richer individuals might demand overprovision.	Good
Moreno-Maldonado, C. et al., (2018) [55]	To analyze the relationships among classic socioeconomic indicators (education and occupation) and others that have been proposed more recently (family affluence scale and subjective family wealth).	Cross-sectional	Adolescents aged between 11 and 16 years, *n* = 8739 (4504 girls and 4235 boys)	2014	Heath-related quality of life, self-rated health, psychosomatic complaints	Parental occupation, parental education, family material wealth, perceived family wealth, life satisfaction	The results highlighted the importance of including different indicators for measuring socioeconomic inequalities in adolescent health.	Good
Arias-de la Torre, J. et al., (2019) [56]	To document the prevalence of poor mental health by gender and social class, and to analyze if poor mental health is associated with the family roles or the employment status inside and outside the household.	Cross-sectional	People aged between 16 and 65 years, *n* = 14,247 (50.6% women)	July 2011 and June 2012	Mental health	Occupational social class, employment status, family and household characteristics, and education	The results of the current study show that, in Spain, there are still gender and social class differences in mental health.	Average
Puig-Barrachina et al., (2011) [57]	To investigate the impact of unemployment on mental health outcomes among vulnerable groups.	Cross-sectional	Population aged between 25 and 64 years, *n* = 8591 (4820 men and 3695 women)	2006	Mental health status	Employment status, social class	Primary findings indicate that unemployment has a greater adverse effect on the mental health of male manual workers, single mothers, main-earner women, and manual workers without unemployment benefits for both sexes.	Good
Clemente López, J. et al., (2019) [58]	To analyze socioeconomic inequalities in mortality in Spain considering geographical and labor market effects on the mortality rate.	Cross-sectional	All the individuals who have had any relationship with social security in the reference year	From 2007 to 2009	Standardized probability of death	Contribution group (social class), educational level, place of residence	The importance of geographical, economic factors in the mortality rate of workers, and the relevance of considering them when forming policies, such as policies related to retirement age.	Average
Bilal, U. et al., (2019) [59]	To explore the association between neighborhood social and economic change and type 2 diabetes incidence in Madrid, Spain, using an electronic health records-based cohort of 200,000 people followed for 6 years.	Cohort	Population living in the city of Madrid, *n* = 199,621 (56.3% women)	2009–2014	Type 2 diabetes incidence	Percentage of foreign born (non-OECD), unemployment, mobility throughput (young/non-OECD), diversity (country of origin), property value, housing renovations, education, total population, new housing, age, mobility throughput, diversity (education/country of origin)	Adjusted results showed an association between neighborhood change and diabetes incidence: compared to those living in aging/stable areas, people living in declining socioeconomic status (SES), new housing and improving SES areas have an 8% (HR = 0.92, 95% CI 0.87–0.99), 9% (HR = 0.91, 95% CI 0.81–1.01), and 11% (HR = 0.89, 95% CI 0.81–0.98) decrease in diabetes incidence.	Good
Orueta, J.F. et al., (2013) [60]	To present an overview of the prevalence of multimorbidity by deprivation level in the elderly population of the Basque Country.	Cross-sectional	Inhabitants of the Basque Country aged 65 years and over, *n* = 452,698 (192,558 men and 260,140 women)	2013	Multimorbidity (to be the co-occurrence of two or more health problems in the same person)	Deprivation index	Multimorbidity is very common among people over 65 years old in the Basque Country, particularly in unfavourable socioeconomic environments.	Good
Belzunegui-Eraso, A. et al., (2018) [61]	To analyze the possible association between health and poverty and social exclusion by comparing populations with and without disability.	Case-control	Population with a disability recognized by a certificate greater than 33 percent, *n* = 14,614 (6659 men and 7955 women)	2012	Severe difficulties for activities daily living (ADL)	Income, type of household, difficulties in meeting certain expenses, social exclusion determinants index (synthetic index)	There is a link between disability and the risk of social exclusion. This relationship is accompanied by other factors such as having severe limitations, suffering a chronic disease, being a woman.	Average
Esteban-Peña, M.M. et al., (2020) [30]	To determine the level of association between territory and vulnerability, specifying proposals for territorial intervention through key socio-health indicators.	Cross-sectional	Non-institutional population of the city of Madrid, *n* = 9513	2017	Self-perception of health status, quality of life in relation to health, tobacco consumption, alcohol consumption, self-referential obesity, sedentary, mental health	The vulnerability-ranking index (immigrant rate, average life expectancy at birth, level of education, number of inhabitants, average income per household, absolute unemployment rate, unemployment rate for over 45 years, unemployment rate for persons without benefits, cadastral value of dwellings, demand rate for dependents, families with a minimum income, and a Home Help Service fee for dependents and a telecare fee for dependents)	There are health problems that are clearly linked to the territory.	Average
Pedrós Barnils, N. et al., (2020) [62]	To (1) explore how intersectional social positions of social class, gender, and region are reflected in population patterns of SRH among Spanish adults, and (2) to examine the contribution of intermediary social processes material and psychosocial factors to these inequalities in self-rated health (SRH).	Cross-sectional	All non-institutionalized people residing in Spain aged 16 years old or older as of 31 December 2014, *n* = 22,456 (11,376 men and 11,080 women)	March–July 2015	Self-rated health (SRH)	Social class (manual or non-manual) and regional development (high or low)	Inequalities in the intersection of social class and regional development were best explained by the joint contributions of material and psychosocial factors, while gender inequalities within non-manual social class were better explained by material factors alone.	Good
Garrido-Cumbrera, M. et al., (2010) [63]	To assess social class inequalities in health services utilization in Spain during 2006.	Cross-sectional	Spanish population aged 16 years or older, non-institutionalized, *n* = 29,478 (14,459 men and 15,019 women)	From June 2006 to June 2007	Visit to a general practitioner (GP) in the 4 weeks before the interview, visit to a specialist in the 4 weeks before the interview, visit to a dentist in the past 3 months, contact with emergency services in the past year, hospitalization in the past year, waiting time for the last visit to a GP or specialist, breast cancer screening of women aged 50 to 69 years in the past 2 years, cervical cancer screening of women aged from 25 to 65 years in the past 3 years, seasonal influenza vaccination of people above 60 years of age in the past year, and self-assessed general health	Social class, health insurance coverage	The results of this study show that, although Spain now has a National Health System, social inequalities still exist in the use of some curative and preventive services.	Average
Amengual-Moreno, M. et al., (2020) [64]	To determine the relation of social determinants in the incidence of COVID-19 in the city of Barcelona.	Ecological	Total population of Barcelona	2020	Total cases of COVID-19 confirmed by protein C reactive (PCR) test, cumulative incidence, smokers, people with body mass index (BMI) > 25 (%), people with one or more comorbidities (%)	Available income per family index	The social determinants correlate with a modification of the incidence of COVID-19 in the neighborhoods of Barcelona, with particular relevance of the prevalence of BMI > 25 and the percentage of immigrants and their origin.	Good
Terán, J.M. et al., (2018) [65]	To assess the impact of the economic crisis on disparities in the prevalence and risk of low birth weight (LBW) according to the maternal socioeconomic profile.	Cross-sectional	Mothers with Spanish nationality with births in the years indicated, *n* = 1779, 506 (all women)	Years 2007, 2009, 2011, 2013, and 2015	Number of live births (primiparous and multiparous), weeks of gestation, birth weight, type of delivery.	Maternal occupation, maternal educational level, marital status, place of residence	The results confirm the persistence of social inequalities in perinatal health described before the crisis, as well as the negative effect of the recession in the period 2007–2015. The results also confirm that disparities in BPN are more clearly associated with the educational level of mothers than with their occupation.	Good
Hurtado, J.L. et al., (2015) [66]	To describe the magnitude of social inequalities in population-based Colorectal Cancer (CRC) screening in the Basque Country between 2009 and 2011, according to the level of socioeconomic deprivation of the area of residence, focusing mainly on response rates and lesions identified.	Cross-sectional	People from the Basque Country between 50 and 69 years of age invited to participate for the first time in colorectal cancer screening, *n* = 219,120 (50.9% women)	2009–2011	Screening participants, FIT test result	Rate of deprivation (dividing population by quintiles)	Sex and socioeconomic group influence the rate of participation in the CRC program and the rate of lesions found in the participants.	Good
Haeberer, M. et al., (2020) [67]	To carry out a comprehensive assessment of social inequalities in cardiovascular disease (CVD) mortality in Spain, in 2015, from an intersectional perspective, considering the combined influence of sex, age, and educational level.	Cross-sectional	All deaths by CVD in the population aged 30 and over in Spain	2015	Total CVD mortality, ischemic heart disease, heart failure, and cerebrovascular disease	Educational level	Cardiovascular mortality is inversely associated with educational level.	Good
Pérez-Hernández, B. et al., (2017) [68]	To examine the distribution of the main cardiovascular risk factors (CVRF) according to socioeconomic level (SEL) among older adults in Spain.	Cross-sectional	Spanish population over 60 years not institutionalized, *n* = 2699 (53% female)	2008–2010	Tobacco and alcohol consumption, adherence to the Mediterranean diet, physical activity and sedentary, weight, waist size and circumference, blood pressure, hypercholesterolemia, glycaemia	Educational level, occupations of subject and parent	There are significant social inequalities in CVRF among older adults in Spain. Reducing these inequalities, bringing the levels of CVRF in those from lower SEL in line with the levels seen in higher SEL, could substantially reduce the prevalence of CVRF in the older adult population.	
Forcadell-Díez, L. et al., (2020) [69]	To describe social inequalities in fertility patterns among women who gave birth between 2007 and 2016 in the city of Barcelona (Spain) by jointly evaluating the effect of individual and socioeconomic neighborhood characteristics.	Cross-sectional	Women aged 15 to 49 years residing in the city Barcelona from 2007 to 2016, *n* = 3,889,396 (all women)	2007–2016	Fertility rate (FR)	Women’s educational attainment, women’s country of origin, classified into three groups, disposable household income (DHI) per capita, percentage of unemployment	The neighborhood’s characteristics played an important role in fertility patterns, independently of women’s individual characteristics.	Good
Duarte-Salles, T. et al., (2011) [70]	To describe social inequalities in obesity and other health problems among adolescents, by sex.	Cross-sectional	Barcelona teenagers between 12 and 16 years of age, *n* = 903 (52% boys and 48% girls)	April and June 2006	Self-perceived health, Body mass index (BMI), chronic diseases or disabilities	Social class, highest level of education of both parents, the family affluence scale (FAS–SES), family structure	This study has shown that there are social inequalities among adolescents, whereby those from a less privileged socioeconomic position have a higher probability of presenting worse health indicators and a lower probability of reporting very good perceived health status.	Good
Urbanos-Garrido, R.M. (2012) [71]	To provide new evidence about the factors driving socioeconomic inequalities in health for the Spanish population by including housing deprivation and social interactions as health determinants.	Cross-sectional	Residents of Spain over 16 years old, *n* = 25,498	2006	Self-assessed health, the presence of chronic diseases, and the presence of conditions limiting daily activities	Educational level, labor status, equivalent household income, financial deprivation	Although mean health levels in Spain are high, socioeconomic inequalities in health are statistically significant.	Good
Barriuso-Lapresa, L. et al., (2012) [72]	To assess mental health and health-related quality of life (HRQoL) of children and adolescents in Spain and to investigate the existence of a social gradient in mental health and HRQOL.	Cross-sectional	Population aged from 4 to 15 years, living in Spain, *n* = 6414 children and young adolescents (3274 boys and 3139 girls)	2006	Mental health, health and health-related quality of life	Place of residence, place of birth, and family socioeconomic status with regard to social class and maternal educational level	There is a social gradient in the mental health of children and young adolescents in Spain. No social gradient was found for HRQoL.	Good
Hernández-Yumar, A. et al., (2018) [73]	To investigate socioeconomic differences in the body mass index (BMI) in Spain.	Cross-sectional	Population living in Spain over 18 years of age during the study, *n* = 14,190 (51.93% female)	2011–2012	Self-reported BMI	Income (ten bands defined by the Spanish National Health Survey), education (nine levels), household composition (living alone and cohabiting)	We confirm the existence of a socioeconomic gradient in BMI. Underlying the intersectional strata heterogeneity, we observed that differences in BMI are particularly notable across levels of educational achievement, especially in women.	Good
Ibáñez, B. et al., (2018) [74]	To determine if the achievement of control targets in patients with type 2 diabetes was associated with personal socioeconomic factors and if these associations were sex dependent.	Cross-sectional	Diabetes population, *n* = 32,638 (56% men)	2014	Glycated hemoglobin (HbA1c) level, systolic and diastolic blood pressure, LDL- and HDL-cholesterol and triglyceride levels, weight, height, body mass index (BMI), smoking status, medication use	Educational and copayment level (socioeconomic status)	We found the presence of socioeconomic inequalities in the achievement of control targets in type 2 diabetic patients.	Good
Marí-Dell’Olmo, M. et al., (2021) [75]	To analyze social inequalities in COVID-19 incidence, stratified by age, sex, geographical area, and income in Barcelona, during the first two waves of the pandemic	Ecological	Non-institutionalized population of Barcelona residents, *n* = 61,572	1 March to 15 July and 16 July to 30 November 2020	Incidence of COVID-19 per 100,000 inhabitants	Income index categorized into quintiles	The results indicate the existence of social inequalities in the incidence of COVID19 by age group, gender, geographical area, and income.	Average
Alberto Capurro, D. et al., (2017) [76]	To describe socioeconomic inequalities in dental health among Spanish middle-aged adults, and the role of behavioral and psychosocial factors in explaining these inequalities.	Cross-sectional	The entire population of Spain, *n* = 17,602 (50.1% male and 49.9% female)	2006	Dental health status based on self-reported dental problems, smoking, frequency of sweet consumption, frequency of sugar-sweetened beverage consumption, frequency of tooth brushing, and dental care utilization	Education, income, and occupational class; psychological distress; perceived family functioning; social support; work-related stress; job satisfaction	This study shows socioeconomic inequalities in dental health among middle-aged adults in Spain.	Good
Larrañaga, I. et al., (2013) [77]	To analyze social inequalities among pregnant women from four cohorts.	Cohort	Pregnant women from Valencia, Sabadell (Catalonia), Asturias, and Gipuzkoa, *n* = 2607 pregnant women	November 2003 and January 2008	Physical activity during the first trimester, smoking status, dietary intake, maternal health (fever, urinary tract infection, hypertensive disorders, …), hours of sleep, planned pregnancy, number of prenatal appointments, pre-pregnancy body mass index (BMI), and weight gain during pregnancy	Occupation and education	Our results highlight the existence of inequalities in self-care during pregnancy and in certain habits and behaviors that have an impact on the health of the mother and child, with pregnant women from the less advantaged social classes having less healthy habits.	Good
Clara Zoni, A. et al., (2017) [78]	To analyze the injuries treated in primary care (wounds, bruises, sprains, fractures, burns, injuries due to foreign body, and poisonings) in the community of Madrid, Spain, by SES, sex, and age.	Cross-sectional	Total population of the community of Madrid, *n* = 6,353,388 (3,306,154 women and 3,047,234 men)	2012	Episodes coded as having an ”injury component” in the PCEMR (fractures, sprains, wounds, burns, foreign body injuries, poisoning, and bruises)	Socioeconomic status (deprivation index)	People with lower SES from the community of Madrid were at a greater risk of injury.	Good
Bilal, U. et al., (2019) [79]	To evaluate the associations among individual-level SES, life expectancy, and mortality of adult men and women from the general population living in a universal healthcare coverage setting.	Cross-sectional	The entire population of Catalonia, *n* = 6,027,424 (2,943,265 men and 3,084,159 women)	2016	Life expectancy, age-adjusted mortality (Catalan Health Surveillance System)	Annual income (SES)	Low SES individuals have much lower life expectancy and higher mortality.	Good
Orueta, J.F. et al., (2013) [80]	To measure the level of socioeconomic-related inequality in the prevalence of chronic diseases and to investigate the extent and direction of inequities in health care provision.	Cross-sectional	Every individual who on 31 August 2011 was covered by public health insurance in the Basque Country and who had been covered for at least 6 months in the previous year, *n* = 2,262,686 (50.9% female)	From September 2007 to August 2011	Multimorbidity (to be the co-occurrence of two or more health problems in the same person)	Deprivation index	Although the relationship between low socioeconomic status and poorer health status is complex and the underlying mechanisms have not been clearly established, the existence of health inequalities is well recognized.	Good
Zapata-Moya, A.R. et al., (2019) [81]	To elaborate on the fundamental cause theory (FCT) by applying the theoretical principles of the DOI theory to preventive healthcare use.	Cross-sectional	Non-institutionalized adults between 50 and 69 years of age, *n* = 7938	2010	Fecal occult blood (FOB) tests (used for colorectal cancer screening), prostate specific antigen (PSA) tests, Papanicolaou test (cervical cancer), mammograms, cholesterol readings, blood pressure checks	Socioeconomic status (SES)	The findings that we have reported suggest that, while social inequalities in specific preventive care practices tend to arise and disappear along the continuing process of technological innovations in cancer screening, they tend to be reproduced overall.	Good
Pascual-Sáez, M. et al., (2019) [82]	To analyze trends in individual health status in Spain.	Ecological	The entire population of Spain	2008–2016	Self-assessed health	Income, social exclusion, household conditions, poverty, education, work, Foster–Greer–Thorbecke (FGT) index	The results show a negative growth if a poor SAH status is chosen as a health poverty threshold, and a growth of health poverty, if a fair self-assessed health status is chosen, revealing a rise of health poverty in Spain.	
Bartoll, X. et al., (2014) [83]	To evidence the recession’s effects on mental health outcomes in Spain.	Cross-sectional	Spanish people aged 16–64 years 2006/2007: *n* = 23,760 (12,019 men and 11,741 women) 2011/2012: *n* = 16,616 (8355 men and 8261 women).	2006–2007 and 2011–2012	Mental health	Social class and level of education	Socioeconomic inequalities in the prevalence of mental health increased among men, but remained stable among women.	Good
Calzón Fernández, S. et al., (2015) [84]	To analyze the impact that the economic crisis and the evolution of socioeconomic inequality before (2007) and during (2011) the current crisis have had on unmet dental care needs in Spain.	Cross-sectional	Spanish population under 65 years of age, *n* = 44,138 (48.8% men and 51.2% women)	2007–2011	Prevalence of unmet dental care, presence of chronic disease	Educational level, marital status, employment status, and income level	There has been an increase in unmet dental care needs as well as in the social gradient for service access. The economic crisis has also increased this unmet need.	Good
de Bont, J. et al., (2020) [85]	To examine how time trends in the prevalence and incidence of overweight and obesity among children and adolescents differ by age, sex, socioeconomic status, urban/rural residence, and nationality.	Cohort	Children and adolescents between the ages of 2 and 17 years, *n* = 1,166,609 (48.9% girls)	2016 and 2018	Overweight and obesity	Deprivation index, nationality, urban/rural residence	We identified sociodemographic groups, including children living in the most deprived areas and with non-Spanish nationalities, among whom prevalence increased, giving rise to increasing deprivation disparities in childhood obesity.	Good

## Data Availability

All necessary data are supplied and available in the manuscript; however, the corresponding author will provide the dataset upon request.

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
