# Peer review of "Socioeconomic Inequalities as a Cause of Health Inequities in Spain: A Scoping Review"

_healthcare, 2023, doi:10.3390/healthcare11233035_

Round 1

Reviewer 1 Report

Comments and Suggestions for Authors

The topic of this review is important and urgent, and the methodology seems appropriate. This review is detailed in several respects while other information is unclear.

It should be made clearer that the focus of the review is not on summarizing the study results in terms of the effects of socioeconomic inequality and health found in the included studies – at least, this is how I understood the purpose of the review as stated by the authors. In its current form, one could get the impression that relevant (partial) results are missing. This should be addressed in a revision of the paper.

Keywords

I would suggest adding “scoping review” as another keyword.

Introduction

p. 1, line 34: Please briefly outline the characteristics of universal health care systems.

p. 1, lines 37 et seq.: The introduction might benefit from a shorter presentation (e.g., illustration of social class from a Marxist perspective).

p. 2, lines 80 et seq.: I would suggest presenting some specifics or differences in theories described by the aforementioned synthesis (e.g., causes, consequences, focus, level of analysis such as macro level of health care) rather than naming some authors. This would be more informative in my view.

p. 2, line 82: Something seems to be wrong in this sentence, please check and revise (“once” instead of “one’s”?).

p. 2, lines 85 et seq.: The transition between the prior paragraph and this one dealing with the Spanish health care system isn’t coherent in my view: How does the information on the Spanish health care system relate to the explanations from the prior paragraph? Was the Spanish system used as an example of a universal health care system? I assume so, but this should be explained more clearly, also with regard to the following research questions.

Method

2.1 Design

p. 3, line 106: Please write “design” instead of “desing”.

2.2. Phase I…

p. 3: Did the authors apply the PCC scheme recommended for scoping reviews (see Peters et al. (2020). Updated methodological guidance for the conduct of scoping reviews. JBI Evidence Synthesis, 18, 2119-2126)? If so, which elements were included as population, concept, and context?

2.3. Phase II…

p. 3: I would suggest presenting the information on the databases used first, followed by the general search strategy (search strings, table 1).

2.6. Phase V…

p. 4, line 167: What is meant by “territorial framework”? Please explain.

Results

p. 4, lines 170 et seq.: It would be helpful to refer to the flowchart (figure 1) and the explanations given there from the outset throughout the text. For instance, it should be added that n=278 hits were eliminated because they were duplicates etc.

p. 4, line 187: Please delete “in”.

p. 5, line 197: Please revise the figure caption.

p. 5, line 198: Please add “included” in the sentence to make clear that the following statements refer to the articles eventually included in the review.

p. 5, line 206: The percentages given here do not add up to 100%. Please check if necessary.

p. 6, table 2: What is meant by ABVD? Please explain this abbreviation.

p. 7, table 3: Please check the page numbering, this page is shown as p. 1 of 34.

table 3: Various abbreviations are used without explanation (e.g., IP (Compes Dea et al.), AP (Aguilar-Palacio et al.), CHF (Cainzos-Achirica et al.)). Also, please explain the meaning of the smiley categories in the “quality” column as a note.

p. 17 (Garcia-Goni et al.): Please write “rehabilitation” instead of “rhabilitation”.

While table 3 is very detailed, what is missing in my view is a compact summary of the findings, if necessary in graphic form. The information in the table is very extensive, readers may find it a bit overwhelming. They will probably find a summary with key information helpful.

Discussion

p. 31: The same is true in my view for the start of the discussion section. I would expect to find some statements on the main findings first. What is the “take home message”/essence of the study results? It remains a bit unclear whether the goal of the review was to present both results on health and socioeconomic indicators etc., and “mechanisms” of their associations, outcomes etc.. The review in general and the discussion section would benefit from clarifying this issue in my view.

p. 31, line 262: Similarly, if the authors state that the data are “quite clear”, wouldn’t it be useful to learn about what these data tell us? Otherwise, the conclusions drawn (p. 32) are somewhat limited.

p. 31, lines 233 et seq.: I would suggest shortening the first paragraph to something like “there is an increase in research on social inequalities and their effects on health/health inequities, coming from various perspectives” or similar.

p. 31, lines 281 et seq.: Please revise this sentence (“which” can be deleted).

Again, please check the page numbering of the discussion section, it starts as p. 1 of 34.

References/Results

For reasons of transparency and completeness, I suggest to add the publications included in the review as references either in the “references” section or in a supplemental file.

Author Response

Dear Reviewer, We sincerely would like to appreciate the comments and contributions made on our work. We are flattered by the review you have made. We really think that they helped us improve the quality of the presentation of our results and clarify methodological aspects that were not clear. You will find a document with the comments that includes a brief response/justification for the comment made and the page where the change was made, when necessary. We hope that the adjustments made are to your liking and meet your expectations for the article to be published. Sincerely, The research team.

Reviewer 2 Report

Comments and Suggestions for Authors

Peer-review report

I sincerely thank you for this opportunity to peer-review the article entitled "Socioeconomic Inequalities as a Cause of Health Inequities in Spain: A Scoping Review." This review has importance in its field. It has been organized well and written scientifically. 

Abstract

Lines 17-18: I suggest the authors include the following: How the included article quality was determined? Followed by “A total of 58 articles consisting n=population with an average age of xxx were included.” 

Keyword: I recommend including "public health." Other terms that are not in the title of this review are also present.

Introduction

Line 98-104: I suggest the authors simplify the objectives of this review.

Method

Lines 127-128: What was the reason that authors considered the three electronic databases? Please specify.

Line 130: Is it two authors? If yes provide their initials like “Two authors (x.x. and x.x.)….”

Line 136: Is it 2004 or 2003? Check in the abstract at line 16.

Line 148: Who was read independently? Can you provide the initials of the authors of this review? For example, “First, x.x., x.x., and x.x. reviewer independently read…”

Lines 1454-155: What are the acceptable levels of methodological quality? Please justify. How did authors determine methodological quality? For example, excellent, good, poor, etc… and related cut-off scores.

Line 161: Who are the two researchers?

Lise 164: What are the descriptive statistics were analyzed? How the authors determined the results of Table 2. Please elaborate in the Phase V section.

Line 173: Who are the two researchers?

Results

Line 192: What do the minimum quality standards mean? Have the authors provided these details in the method section at the appropriate place?

Line 197: I suggest the authors modify the Figure 1 title. What is the “Population size that excluded 10 articles? Is it less, more, or inadequate to a certain size? Please provide more details.

Table 3: The authors' column should be “Authors (Year).” In the population column, include the population size in each study specifying their gender or sex. What do symbols in a quality column represent?

Discussion

What were the key results? Please summarize first and discuss it later.

Author Response

(The authors gave the same response as above.)

Round 2

Reviewer 1 Report

Comments and Suggestions for Authors

First of all, I would like to thank the authors for providing a thorough revision of their paper. My comments and suggestions were adequately addressed. I would like to point out some minor spelling/wording issues that should be corrected prior to publication:

Abstract

p. 1, line 18: Please write either “…carried out in the Spanish healthcare system” or simply “…in Spain”, as it was before the correction.

p. 1, line 23: Please write “were” instead of “where”.

Introduction

p. 1, lines 40 et seq.: Thank you for adding this information. But please have a second look at the sentence. Did you mean “In this type of national health systems, the financing comes from the state budgets…”?

p. 2, line 58: Please write “this map” instead of “these…”.

p. 2, lines 95-96: I don’t understand the meaning of the sentence that was added in the review – please check again.

p. 3, line 99: Please delete “fact” in this sentence.

Results

p. 8, lines 271-272: Two minor comments – please write “…list of references…” and “…can be found in table 3…”.

Discussion

p. 34 (?), lines 301 et seq.: This sentence (“Also, the method…”) is difficult to understand, please check.

Author Response

Dear Reviewer,

We appreciate your feedback regarding the changes made to our work during the first round of review. We also want to thank you for the suggestions made in this second round.

In this document, you will find a table with the changes made. These changes were marked in blue in the text to differentiate them from the changes in the first round.

It is a pleasure to work with people who are involved and attentive to details.

We are looking forward to your response.

Sincerely,

The research team.
